# Characterization of Photocurable IP-PDMS for Soft Micro Systems Fabricated by Two-Photon Polymerization 3D Printing

**DOI:** 10.3390/polym15224377

**Published:** 2023-11-10

**Authors:** Rishikesh Srinivasaraghavan Govindarajan, Stanislav Sikulskyi, Zefu Ren, Taylor Stark, Daewon Kim

**Affiliations:** Department of Aerospace Engineering, Embry-Riddle Aeronautical University, Daytona Beach, FL 32114, USA; srinivr1@my.erau.edu (R.S.G.); sikulskyi@gmail.com (S.S.); renz@my.erau.edu (Z.R.); starkt1@my.erau.edu (T.S.)

**Keywords:** two-photon polymerization (2PP), 3D printing, microfabrication, IP-PDMS, material characterization, nanoindentation, Young’s modulus, DMA, creep, dielectric properties

## Abstract

Recent developments in micro-scale additive manufacturing (AM) have opened new possibilities in state-of-the-art areas, including microelectromechanical systems (MEMS) with intrinsically soft and compliant components. While fabrication with soft materials further complicates micro-scale AM, a soft photocurable polydimethylsiloxane (PDMS) resin, IP-PDMS, has recently entered the market of two-photon polymerization (2PP) AM. To facilitate the development of microdevices with soft components through the application of 2PP technique and IP-PDMS material, this research paper presents a comprehensive material characterization of IP-PDMS. The significance of this study lies in the scarcity of existing research on this material and the thorough investigation of its properties, many of which are reported here for the first time. Particularly, for uncured IP-PDMS resin, this work evaluates a surface tension of 26.7 ± 4.2 mN/m, a contact angle with glass of 11.5 ± 0.6°, spin-coating behavior, a transmittance of more than 90% above 440 nm wavelength, and FTIR with all the properties reported for the first time. For cured IP-PDMS, novel characterizations include a small mechanical creep, a velocity-dependent friction coefficient with glass, a typical dielectric permittivity value of 2.63 ± 0.02, a high dielectric/breakdown strength for 3D-printed elastomers of up to 73.3 ± 13.3 V/µm and typical values for a spin coated elastomer of 85.7 ± 12.4 V/µm, while the measured contact angle with water of 103.7 ± 0.5°, Young’s modulus of 5.96 ± 0.2 MPa, and viscoelastic DMA mechanical characterization are compared with the previously reported values. Friction, permittivity, contact angle with water, and some of the breakdown strength measurements were performed with spin-coated cured IP-PDMS samples. Based on the performed characterization, IP-PDMS shows itself to be a promising material for micro-scale soft MEMS, including microfluidics, storage devices, and micro-scale smart material technologies.

## 1. Introduction

Recent developments in micro-scale additive manufacturing (AM) open new possibilities in state-of-the-art areas such as microelectromechanical systems (MEMS), nanotechnology, microfluidics, lab-on-a-chip, etc. [1,2]. While changes at the micro/nano-scale alter the behavior of fabricated objects due to the nonclassical physics of their components and materials, AM reveals the true potential of complex microstructures and micro-scale designs. Furthermore, fabricating devices with soft elements can enable new functionalities and the incorporation of smart materials, the performance of which is not limited and often even improved by miniaturization [3,4]. While achieving moderate flexibility is possible even with stiff materials through the utilization of compliant hinges, soft materials are necessary for structures to possess intrinsic compliance and stretchability. Thus, combining micro-scale AM and soft materials opens a wide range of possible technologies, such as micro-scale soft robotics, soft MEMS, and flexible and stretchable microelectronics [3,5].

Within micro-scale AM, two-photon polymerization (2PP) is of great interest thanks to its resolution, quality, and three-dimensional printing capability [6]. Recently, a new soft material product entered the market of microfabrication through the 2PP process, a biocompatible polydimethylsiloxane (PDMS) photoresin called IP-PDMS, enabling freeform 3D printing of micro-scale elastomeric designs. Besides allowing for more complex designs, 3D printing of soft micro-scale designs replaces time-consuming and limited multi-step molding processes that used 2PP printed molds from stiffer photocurable polymers [7,8].

The manufacturing data for the material report a Young’s modulus of 15.3 MPa and stretchability of at least 240%, which makes the material attractive for micro-scale soft robotics, MEMS, microfluidics, and life sciences [9]. However, due to the recentness of the material and the complicated characterization of small 2PP-printed samples, IP-PDMS’s material characterization in the literature is scarce. Particularly, the first paper that utilized IP-PDMS reported some printing parameters and the basic chemical composition [6]. A few recent studies have investigated the effects of 2PP printing speed, laser power, and ultraviolet-bleaching on the Young’s modulus of fabricated IP-PDMS samples [10,11]. Meanwhile, other studies that utilized IP-PDMS focusing on unique applications reported fascinating designs and functionalities of the produced structures and mechanisms [12,13,14,15,16,17]. The only reported material property within these works was the contact angle of water, ethanol, and isopropanol on 2PP-printed IP-PDMS surface, with the angle value matched for water in multiple studies [13,14]. While considerable results were achieved in the mentioned studies, the studies used empirical approaches for the design and fabrication of the structures due to the lack of available material properties. Restrained modeling capabilities due to the absence of various material properties limit the scientific toolset available to implement the novel soft material. Thus, to facilitate application of IP-PDMS, this paper presents a comprehensive material characterization and investigates the material properties of interest for emerging fields within soft micro-scale devices. Firstly, uncured resin is analyzed using FTIR and compared to common PDMS compositions, such as Sylgard 184 and 186, to emphasize its structural and compositional differences. Secondly, the transmittance, surface tension, and contact angle of uncured resin on glass are analyzed, providing critical data for applications that either have residual uncured resin, require its removal from microfluidic channels, or use it in its uncured state, underlining the significance of its rheological properties. Furthermore, a spin-coating procedure is established that defines the relationship between spin coating parameters and the thickness of the IP-PDMS film, facilitating sample preparation and optimizing material utilization. Additionally, the mechanical properties of the cured resin, such as the static Young’s modulus, DMA viscoelastic properties, and creep, are analyzed using nanoindentation techniques. These measured properties provide insights into the mechanics of soft IP-PDMS material for any application that requires soft and compliant components, including those where a load or deformation is applied dynamically or over an extended period of time. These include micro-size soft actuators, sensors, stretchable devices, electronic skin, and pre-stretched and inflated structures. As areas such as smart micro-adhesion require an understanding of material tribology, a preliminary assessment of both static and dynamic friction coefficients with glass is performed. Lastly, a range of smart material technologies for sensing, actuation, energy storage, and harvesting applications rely on the material’s dielectric properties. Thus, the relative dielectric permittivity and dielectric breakdown (BD) strength of cured IP-PDMS were investigated for both 2PP and spin-coated films.

To the authors’ knowledge, all of the above properties of IP-PDMS, except for the Young’s modulus and some DMA results, are reported in this paper for the first time. Finally, due to the scarcity of existing research on IP-PDMS’s properties, characterization is performed for the commercial IP-PDMS resin as available, e.g., with no material alterations, and for the samples fabricated using recommended 2PP printing parameters.

## 2. Materials and Methods

The main material of the study is a biocompatible elastomeric photoresin called IP-PDMS (Nanoscribe GmbH & Co. KG, Eggenstein-Leopoldshafen, Germany). While limited information is available about the material, some of its main components are mentioned in the safety data sheet. As per its safety data sheet, IP-PDMS contains more than 90 wt.% of (acryloxypropyl)methylsiloxane dimethylsiloxane copolymer, more than 5 wt.% of N,N-Dioctyl-1-octanamine or Trioctylamine, and less than 5 wt.% of (3-acryloxy-2-hydroxypropoxypropyl) terminated PDMS [18], where the structural formulas of all the components are shown in Figure 1. The specified copolymer is the main component of IP-PDMS that cures in the presence of a photoinitiator. The viscosity of the used copolymer can be found to be in the range of 50–125 cP [19]. The used terminated PDMS has low molecular weight of about 600–900 g/mol and, consequently, a low viscosity of about 60–140 cP [19]. Notably, the low viscosity of the above components explains the low viscosity of 100 cP of the final IP-PDMS (uncured) resin as reported by the manufacturer.

Additional materials in the experimentation include PDMS Sylgard 184 (Dow Inc., Midland, MI, USA, part #4019862) and Sylgard 186 (Dow Inc., Midland, MI, USA, part #2099551), both with 10:1 ratios of elastomer base (Part A) to curing agent (Part B), which are used to compare material compositions with IP-PDMS; 99.9% Isopropanol (M.G. Chemicals Ltd., Burlington, ON, Canada, part #824) is also used to perform IP-PDMS development after printing and clean substrates prior to printing, and 99.5% Acetone (Duda Energy LLC, Decatur, AL, USA, part #acetone2) and deionized water of Type III as per ASTM D1193 are used to clean substrates prior to printing.

The following subsections discuss 2PP fabrication, material handling, and testing setups. All material mixing, sample preparation, and testing are performed in a lab environment at a temperature of 21 ± 1 °C and relative humidity of 53 ± 3%.

### 2.1. Sample Preparation

Three sets of IP-PDMS samples were prepared for material characterization. The first set of samples used for nanoindentation were 2PP-printed cylinders with diameters of 400 µm and heights of 100 µm. The selected diameter of 400 µm enables printing the sample without stitching to avoid the potential effect of this process on the material properties. The second set of samples were 2PP-printed with block shear angles of 15° stitched cylinders with diameters of 3350 µm and thicknesses of 20 µm. These larger cylinders were used for dielectric measurements, particularly the breakdown strength considering the stitching effects. Moreover, flatness of the printed and stitched IP-PDMS surface was evaluated based on a gold-sputtered 3350 µm diameter cylinder utilizing optical profilometer Filmetrics Profilm3D (KLA Corporation, Milpitas, CA, USA) with a 50x Nikon objective. Lastly, the third set of samples were spin-coated films of various thicknesses that enabled measuring the breakdown strength with stitching effects eliminated, the dielectric permittivity, as well as additional testing that utilized a larger surface area of the material. Spin-coating IP-PDMS on conductive indium tin oxide (ITO)-coated 1” square glass substrates (MSE Supplies, Tucson, AZ, USA) allows for using the ITO layer as an electrode below the elastomer for dielectric characterization. After spin coating, nitrogen was supplied to overcome oxygen quenching that prevents UV curing on the air/resin interface of IP-PDMS, and the material was then immediately cured with a 10 W 395 nm wavelength UV lamp for 1 min. Fowler IP54 disk micrometer (Newton, MA, USA) was used to measure the thickness of spin-coated films.

### 2.2. The 2PP Printing Parameters

Printing of the material characterization coupons was accomplished using a Nanoscribe Photonic Professional GT2 system equipped with a Zeiss 25x objective lens (Numerical Aperture 0.8). ITO-coated 1” square glass substrates were used and cleaned with deionized water, acetone, and isopropanol in a sonicator and blow-dried with an handheld air bulb blower. The substrates were activated with oxygen plasma using a Plasma Etch PE-50HF system (Carson City, NV, USA). Table 1 lists the parameters used for IP-PDMS printing. To reduce stitching effects while maintaining high print speed, galvo scanning was used for x- and y-printing along with the piezo system for z-movements of printing blocks. The z-drive was used for printing a single block zone that extends past the piezo range (300 µm) while staying away from block stitching effects.

### 2.3. Wettability Tests

A static sessile drop test was performed to determine the contact angle of IP-PDMS (uncured) resin on a glass surface. IP-PDMS droplets of 1 µL were deposited in a static fashion using a micropipette on a clean horizontal glass surface and allowed to reach thermodynamic equilibrium by waiting for one minute. As IP-PDMS does not have solvents in its composition, a longer waiting time does not affect the measured contact angle, which is observed in the test. Pictures of the droplet are captured using an optical microscope Dino-Lite AF4515ZTL (Bangkok, Thailand) and processed to measure the contact angle using ImageJ software.

The second wettability test is a capillary rise test in which the surface tension of IP-PDMS (uncured) resin is determined. Open-ended glass capillary Corning 9530-2 Pyrex tubes (Corning, NY, USA) were lowered into IP-PDMS (uncured) resin. The maximum capillary rise reached in the test and pre-determined contact angle between IP-PDMS (uncured) resin and glass were used to calculate the surface tension of IP-PDMS (uncured) resin.

### 2.4. FTIR

FTIR spectra were recorded using an Agilent Cary 630 spectrometer (Santa Clara, CA, USA). The spectra of unpolymerized IP-PDMS and commercial PDMS (Sylgard 184 and 186: Elastomer base) were collected in the transmission mode with a wavenumber range from 650 to 4000 cm^−1^ at 16 scans per spectrum followed by an automatic background correction at 4 cm^−1^ resolution.

### 2.5. Nanoindentation

Various mechanical characterizations of IP-PDMS were performed utilizing a Bruker Hysitron TI-980 nanoindenter (Billerica, MA, USA) with the 10 mN low load transducer. All nanoindentation tests were performed on 2PP-printed 400 µm diameter cylindrical samples with applied force of 250 µN. For the single indentation test, a 30 s dwell time was used to minimize viscoelastic effect on the measured material’s reduced modulus. Next, nanoscale dynamic mechanical analysis (nanoDMA) was performed with the frequency range between 10 and 220 Hz. Lastly, creep characterization of IP-PDMS was performed over 1 h at a sampling rate of 220 Hz.

### 2.6. Friction Test

A custom setup consisting of a micromanipulator and single axial 50 g S-beam load cell FUTEK LSB200 (Irvine, CA, USA) was used. An ITO-coated glass with spin-coated IP-PDMS film and a surface area of about 1 cm^2^ was fixed on the micromanipulator. Another clean glass with an attached 123 g weight was placed on top of the IP-PDMS film and attached to the load cell through a flexible rope. By moving the micromanipulator at various rates from the resting position, the friction between the glass and IP-PDMS surfaces was initiated, enabling the calculation of the static and dynamic friction coefficients.

### 2.7. Dielectric Properties Tests

Relative dielectric permittivity of IP-PDMS was determined through a capacitance measurement of a 113 µm thick spin-coated film. Once samples were spin-coated and cured, a small piece of IP-PDMS was removed from one corner of the ITO-coated glass to expose the conductive ITO layer. An LCR meter GW Instek precision LCR-6020 (Montclair, CA, USA) was used to measure the capacitance of the IP-PDMS parallel plate capacitor, where ITO served as a bottom electrode and an aluminum circular electrode with diameter of 9.5 mm was placed on top of the IP-PDMS layer. Once the capacitance value was measured, the relative dielectric permittivity was calculated as $\varepsilon_{r}=\left(C/\varepsilon_{0}\right)\left(d/A\right)$, where C is the capacitance, A is the aluminum electrode area, and d is the distance between electrodes (thickness of the tested IP-PDMS film). The fringing field effect on the perimeter of the aluminum electrode was neglected as per DEA testing recommendations ($\sqrt{A}/d\approx 70$) [20]. The same approach and analysis were applied to testing cylindrical coupons that are printed using 2PP on ITO glass.

For the breakdown strength test, characterizing both spin-coated and 2PP-printed coupons is one of the main objectives. The spin-coated coupons had a thickness of 18 and 40 µm while all 2PP-printed samples were maintained at 20 µm. While the ITO layer served as a ground, a 34-gauge wire was bent to a radius of less than 500 µm and served as a positive electrode in the test. A high-voltage amplifier TREK 20/20C-HS (Denver, CO, USA) was used to apply voltage to the positive electrode by the slow rate-of-rise method according to ASTM D149 with an increase rate of 20 V/s until breakdown. The positive wire was handled across the tested film using a WPI HS6 micromanipulator (Sarasota, FL, USA) with a resolution of 5 µm. A microscope, as used in Section 2.3, was used while performing the experiment to monitor the breakdown and further control the contact process between the wire and IP-PDMS film.

## 3. IP-PDMS Characterization

This section demonstrates and discusses the results of various characterizations of IP-PDMS.

As a new material in the field of small-scale fabrication, IP-PDMS underwent material characterization that is important for the design, fabrication, and performance of soft actuators and sensors. Some of the determined material properties are shown in Table 2.

### 3.1. Composition Characterization of Uncured IP-PDMS through FTIR

The infrared absorption spectra of IP-PDMS and commercial PDMS (Sylgard 184 and 186) are obtained to observe the functional group presence. The existence of different functional groups in IP-PDMS can be identified at wavenumbers 794 cm^−1^ (CH_3_ rocking in Si-CH_3_), 1014 and 1043 cm^−1^ (Si-O-Si), 1188 cm^−1^ (-CH_2_-O-CH_2_- stretching vibration), 1258 cm^−1^ (CH_3_ symmetric bending in Si-CH_3_), 1408 cm^−1^ (Si-CH=CH_2_), 1728 cm^−1^ (C=O stretching of acrylate), and 2962 cm^−1^ (C-H stretching in CH_3_), respectively, as shown in Figure 2 [21,22,23]. Additionally, the absorption band intensity change compared to the Sylgard PDMS samples at 1408 cm^−1^, indicating vinyl presence, represents the functional group consumption upon cross-linking reaction in photocurable IP-PDMS resin.

### 3.2. Transmittance of Uncured IP-PDMS

Transmittance of uncured IP-PDMS resin is evaluated within the visible light spectrum for potential applications where uncured IP-PDMS is sealed within a structure or used as a functional material. Figure 3 shows the percentage of transmittance within a 350–700 nm wavelength, demonstrating the high transmittance of uncured IP-PDMS resin at a wavelength around and above 450 nm, comparable with the extensively used PDMS Sylgard 184 (Part A) and higher than PDMS Sylgard 186 (Part A). The considerable absorption of IP-PDMS towards the ultraviolet wavelength region is expected due to the unique 2PP process, where two photons of twice the curing wavelength are near-simultaneously absorbed for polymerization to occur [24]. While using Nanoscribe’s 780 nm wavelength laser, this entails that the curing wavelength of the compatible resins should be around 390 nm.

### 3.3. Wettability Properties of Uncured IP-PDMS

The last set of properties measured for uncured IP-PDMS resin are wettability properties. Firstly, a contact angle of IP-PDMS is determined through the static sessile drop test. Figure 4a demonstrates the equilibrium state of an uncured IP-PDMS droplet, deposited in a static fashion on a clean horizontal glass with the measured contact angle of 11.5 ± 0.6°. The presented standard deviation was calculated utilizing Bessel’s correction for the small sample size of five droplets.

Once the contact angle of IP-PDMS (uncured) resin on glass is determined, a capillary rise test with the glass capillary tube could be used to determine the surface tension of IP-PDMS (uncured) resin. Figure 4b demonstrates the equilibrium capillary rise, $h_{e}$, of 11.1 ± 0.1 mm for IP-PDMS in the given tube, from which the surface tension, γ, of IP-PDMS (uncured) resin is calculated according to Equation (1) [25]
(1)$$h_{e}=\frac{2\gamma cos\left(\theta_{o}\right)}{r\rho g}$$
where $\theta_{o}$ is the equilibrium contact angle between IP-PDMS (uncured) resin on glass, r is the capillary (tube) radius, ρ is the IP-PDMS (uncured) density of 1.01 g/mL, and g is the gravity constant. Considering the capillary tube’s inner diameter of 0.95 ± 0.15 mm, the calculated surface tension of IP-PDMS (uncured) resin was 26.7 ± 4.2 mN/m. The standard deviation of surface tension was calculated considering the propagation of standard deviations of capillary rise ($h_{e}$), contact angle ($\theta_{o}$), and capillary tube radius (r), as independent variables. The high value of the surface tension’s standard deviation is primarily due to the high variation in the capillary tube’s radius. Nevertheless, the knowledge of the material’s surface tension and contact angle on glass is useful for certain applications such as cleaning microfluidic channels. Furthermore, it is also important for predicting the limitations of the 2PP process related to IP-PDMS rheology, e.g., how much IP-PDMS spreads on a glass surface when deposited before the printing process or at which point IP-PDMS separates from the printing lens and wets the printed microstructure. Finally, the contact angle of water on cured IP-PDMS is measured with a spin-coated sample to eliminate the potential effects of 2PP stitching on wettability, as shown in Figure 4c. The measurements showed the values of 101.1 ± 0.6° with a circle fit and 103.7 ± 0.5° with an ellipse fit, close to the previously reported values in a few non-related studies of 108.5 ± 0.44° and 106 ± 6°, where the fit type was not reported [13,14].

### 3.4. Mechanical Properties of Cured IP-PDMS through Nanoindentation

Mechanical properties are amongst most important material response parameters for MEMS application and are therefore thoroughly studied in this paper. Mechanical characterization is performed to evaluate the compliance of the material as a crucial parameter for soft actuator application. Figure 5 demonstrates a force deflection curve obtained through nanoindentation of one of the tested 2PP-printed (cured) IP-PDMS samples. The measured reduced modulus, $Y_{r}$, of the samples are averaged to 7.95 ± 0.26 MPa and converted to the Young’s modulus, Y, value of 5.96 ± 0.20 MPa according to Equation (2), assuming the Poisson’s ratio, ν, of the IP-PDMS to be 0.5. The second component of the equation that contains the Young’s modulus, $Y_{in}$, and Poisson’s ratio, $\nu_{in}$, of the indenter is dropped due to the negligible compliance of the indenter tip ($Y_{in}$ = 1140 GPa, $\nu_{in}$ = 0.07) compared to IP-PDMS.
(2)$$\frac{1}{Y_{r}}=\frac{1-\nu^{2}}{Y}+\frac{1-\nu_{in}^{2}}{Y_{in}}$$

The obtained Young’s modulus value is considerably lower than the 15.3 MPa reported on the manufacturer’s website, despite using the recommended printing parameters for 2PP fabrication of the tested sample and determining both Young’s moduli through nanoindentation [9]. Meanwhile, both values are larger than for the majority of common PDMS compositions in the literature [26,27,28]. Like all elastomers, PDMS is composed of slightly cross-linked polymer chains, and its mechanical properties are sensitive to the degree of polymerization. For example, besides mixing two-component PDMS with a higher fraction of a platinum-based curing agent, curing PDMS at a higher temperature over a sufficient time results in a higher degree of polymerization and stiffness of the material [29]. Therefore, the difference between the manufacturer’s reported value and the value determined in the current study is attributed to possible small adjustments in laser power during the 2PP printing of the IP-PDMS sample. While this paper focuses on the material properties of IP-PDMS fabricated using the recommended printing settings, varying laser powers and other printing parameters are of great interest for future studies to tailor IP-PDMS’s degree of polymerization and resultant material properties.

As Figure 5 shows, the 2PP-printed (cured) IP-PDMS sample is loaded using a nanoindenter tip with a 250 µN force and 30 sec dwell time to minimize the material’s viscoelastic effect on the measured reduced modulus. The test is performed multiple times while increasing the dwell time up to 30 sec, where the measured reduced modulus did not change from the previous measurement by more than 5%. During the 30 s period, a deformation of about 0.2 µm occurred at a constant applied load. Thus, a creep test is performed at room temperature to investigate the material’s creep behavior. As Figure 6 shows, a certain amount of primary creep is present during the first 20 s, and no noticeable secondary creep deformation occurred for 1 h of testing under a constant load. Therefore, it is concluded that IP-PDMS possesses purely viscoelastic rather than viscoplastic behavior at room temperature. That finding agrees with the behavior of conventional PDMS known for its relatively low viscoelastic behavior compared to other material that are used for DEAs [26,30,31,32].

To further characterize the viscoelasticity of 2PP-printed (cured) IP-PDMS, the dynamic mechanical behavior of the material is investigated through nanoscale dynamic mechanical analysis (nanoDMA). Figure 7 demonstrates the storage modulus, loss modulus, and loss factor measured for the frequency range between 10 Hz and 220 Hz. The first noticeable detail from the graph is the higher initial value of the storage modulus than the Young’s modulus determined earlier in this paper. The difference is attributed to the 10 Hz frequency at which the first storage modulus value is measured. As the frequency increases, both storage and loss moduli demonstrate slow and relatively steady growth with low loss factor values. However, a considerable change in behavior is evident near the 180–190 Hz region. Particularly, the storage modulus increases its value and remains at a higher frequency. Meanwhile, the loss modulus and loss factor peak at about 190 Hz and then return to their previous values at lower frequencies. Such a behavior is typical for viscoelastic materials that can be described using the Maxwell representation of the Standard Linear Solid (SLS) model [33,34]. Nonetheless, DMA tests performed on typical temperature-cured PDMS with various degrees of polymerization do not exhibit such a peak across the literature [35,36]. Therefore, it is assumed that this characteristic behavior is introduced to IP-PDMS by adding UV initiators.

### 3.5. Spin Coating of Uncured IP-PDMS

Spin coating of IP-PDMS films is performed for several reasons. Firstly, fabricating films without stitching allows the comparison of breakdown strength with 2PP-printed stitched samples. Secondly, larger areas of the samples increase characterization accuracy for dielectric permittivity and friction coefficient measurements, which improves the testing efficiency. Figure 8 demonstrates the final thicknesses of spin-coated and cured films that are used for measuring dielectric permittivity, breakdown strength, and friction coefficients. When comparing the obtained thickness values for IP-PDMS with the spin-coated thicknesses of common PDMS compositions in the literature, the IP-PDMS thicknesses stand on the lower side of the range [35,37]. This agrees with the authors’ visual observation of IP-PDMS’s lower viscosity compared to commonly used PDMS Sylgard 184.

### 3.6. Friction Coefficients of Cured IP-PDMS

Friction is a critical property that needs to be known for various engineering applications. Despite generally being treated as negative, appropriate friction values are essential to enable many processes based on mechanical contact. Therefore, static and dynamic friction coefficients are determined using a custom setup described in Section 2.6.

The obtained friction coefficients were measured using a custom setup, a schematic of which is shown in Figure 9a, for three different speeds of motion and a single applied normal force, as shown in Figure 9b. The speeds are selected as relevant for the scale of 2PP-fabricated microdevices. To maximize the accuracy of the measurements, the highest possible normal load of 123 g is used such that the measured friction force does not exceed the 50 g capacity of the load cell at the highest tested speed of motion. Both static and dynamic coefficients demonstrate growth as a higher sliding speed is used, with the static friction coefficient being larger for most material pairs. When compared with the values for the dry friction of conventional PDMS in the literature, IP-PDMS’s friction values are slightly lower [38,39]. However, the values found in the literature are determined for considerably higher speeds of motion, whereas the current experiment is designed to serve micro-scale friction devices. Therefore, the direct comparison and validation of results is complicated.

### 3.7. Dielectric Properties of Cured IP-PDMS

Dielectric properties are of great interest for new materials used in the microfabrication of MEMS, as many systems utilize these properties while operating under capacitive or piezoelectric principles or insulating critical elements from applied voltage.

The first conducted dielectric test is a relative permittivity measurement that is performed on 113 µm thick IP-PDMS (spin-coated, cured) films. The measured capacitance is converted to the permittivity value of $\varepsilon_{r}$ = 2.63 ± 0.02, which is within the typical range of relative dielectric permittivity for common PDMS formulations [26,27,28]. An attempt was made to measure the dielectric permittivity based on the 2PP-printed cylindrical coupon. However, the 20 µm thickness of printed coupons limited the accuracy of the measured capacitance due to the imperfect flatness of the top aluminum electrode utilized in the test, resulting in lower values of capacitance and relative permittivity. Despite the absence of a direct comparison, it is believed that stitching does not considerably affect the permittivity of the sample due to their small size.

The second conducted dielectric test is a breakdown strength measurement that is performed on both spin-coated films and 2PP-printed cylindrical samples with comparable thickness. Figure 10a demonstrates the average values and standard deviations achieved in the test for three categories of results. The first value corresponds to the breakdowns that occurred due to the bubbles and other local defects close to the positive electrode in the test. The bubbles are the result of local over-polymerization during 3D printing due to certain defects or impurities in the resin and can clearly be observed in the microscope images (Figure 10b). The second value of dielectric strength is calculated from the breakdown events that happened after the initial breakdown through the nearest defects. This is possible due to the ITO electrode’s self-clearing effect that made the area (affected by the previous breakdown) nonconductive. As a result, the applied voltage could be further increased until the breakdown occurred through the visually defect-free IP-PDMS in the vicinity of the positive electrode. Comparing the dielectric strengths for the first and selected breakdowns, the compromised performance of IP-PDMS with possible fabrication defects is evident. The second objective for the breakdown strength measurements is evaluating the stitching effects. For that reason, the stitching-free, spin-coated IP-PDMS samples are prepared and tested and demonstrate a higher average property value than the 2PP-printed samples. To investigate the cause of the lower breakdown strength in 2PP-printed samples, surface profilometry is performed on the printed cylinders, unveiling local thinning of the film on the stitches of more than 2 µm, as shown in Figure 10c,d. For the 20 µm thick cylinders used in the test, 2 µm thinning could considerably lower the dielectric strength, and the breakdown events were not observed to concentrate where the stitches are but had a more random nature. Thus, it is concluded that the breakdown strength of the material at the stitches is not penalized considerably. Instead, other defects, such as bubbles and impurities, must be minimized to increase the breakdown strength, as is observed from the testing of both 2PP-printed and spin-coated samples. Lastly, an uneven IP-PDMS thickness at the stiches can be a critical factor for very thin defect-free films and needs to be considered. Nevertheless, the measured values for both 2PP-printed and spin-coated samples are well within the PDMS breakdown strength values reported in the literature [40,41,42]. Particularly, 3D-printed silicone films often exhibit low breakdown strength [43]. Meanwhile, 2PP-printed IP-PDMS samples maintained their dielectric strength relatively well and were mostly compromised by the bubbles caused by material over-polymerization. Thus, IP-PDMS can be considered as a promising material for high-voltage applications, where softness might be necessary in the material, such as when used as insulation, dielectric elastomer actuators, high-voltage supercapacitors, as well as some energy harvesting and transmission.

## 4. Conclusions

This study performed a comprehensive material characterization of a new soft material, IP-PDMS, for 2PP micro-scale AM. Based on the obtained characterization results and the current state of the novel material’s utilization in the field, IP-PDMS is considered a promising material to propel state-of-the-art micro-scale fields and enable new capabilities related to the functionalities of soft structures and soft smart-materials-based technologies. Particularly, the material’s room-temperature static modulus of 6 MPa, storage modulus of less than 10 MPa at frequencies below 150 Hz, and low creep allow the utilization of IP-PDMS for micro-scale soft technologies. Regarding soft smart-materials-based technologies, the relative dielectric permittivity of IP-PDMS typical to common silicones allows for IP-PDMS’s application in energy storage, harvesting, sensing, and actuation, where regular PDMS materials can be used. Furthermore, the high breakdown strength of IP-PDMS compared to other 3D-printed PDMS films, achieved through precise dimensional AM and controllable curing through 2PP, extends its capabilities in high-voltage applications. The findings delineated within this work serve as a valuable compass for subsequent utilization of IP-PDMS in different fields of application. Future work on material characterization is encouraged, for instance, investigating DMA at various temperatures, fatigue, and piezoelectric properties, etc.

## Figures and Tables

**Figure 1 polymers-15-04377-f001:**
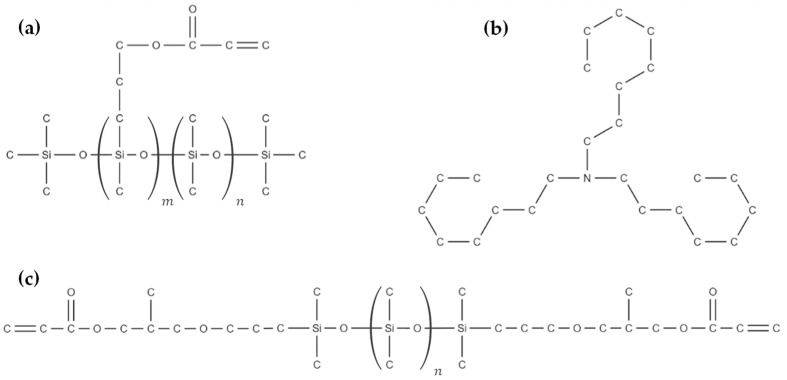
Structural formulas of IP-PDMS components: (**a**) (acryloxypropyl)methylsiloxane dimethylsiloxane copolymer, (**b**) Trioctylamine, and (**c**) (3-acryloxy-2-hydroxypropoxypropyl) terminated PDMS.

**Figure 2 polymers-15-04377-f002:**
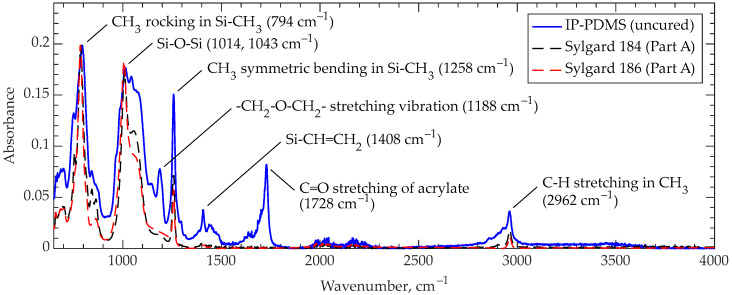
FTIR results for uncured IP-PDMS resin compared with uncured commercial PDMS compositions Sylgard 184 (Part A) and Sylgard 186 (Part A).

**Figure 3 polymers-15-04377-f003:**
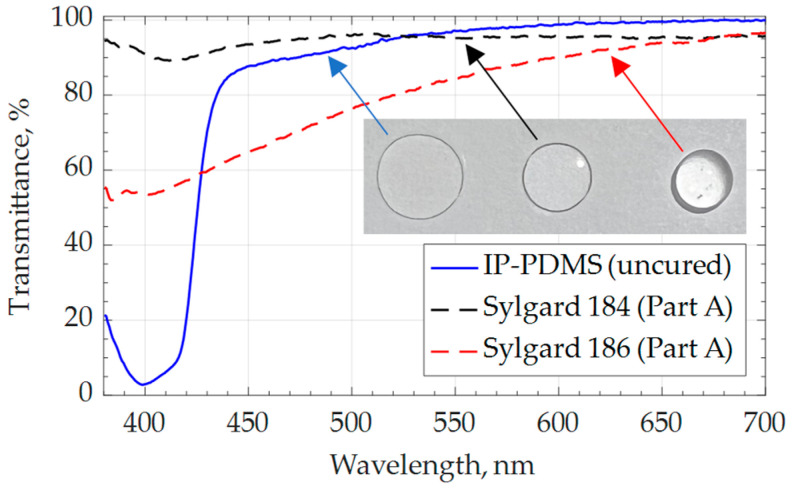
Transmittance of uncured IP-PDMS resin compared with uncured commercial PDMS compositions, Sylgard 184 (Part A) and Sylgard 186 (Part A), within visible wavelength spectrum and visuals of the three materials deposited on a glass surface.

**Figure 4 polymers-15-04377-f004:**
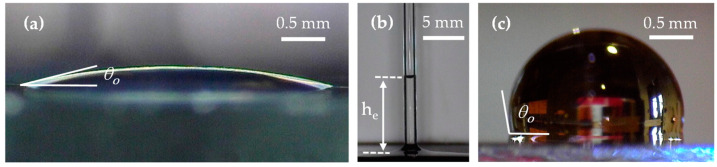
Wetting characterization of IP-PDMS for (**a**) contact angle of uncured IP-PDMS on glass through the static sessile drop test, (**b**) capillary rise of uncured IP-PDMS in a glass capillary tube, and (**c**) contact angle of water on spin-coated (cured) IP-PDMS.

**Figure 5 polymers-15-04377-f005:**
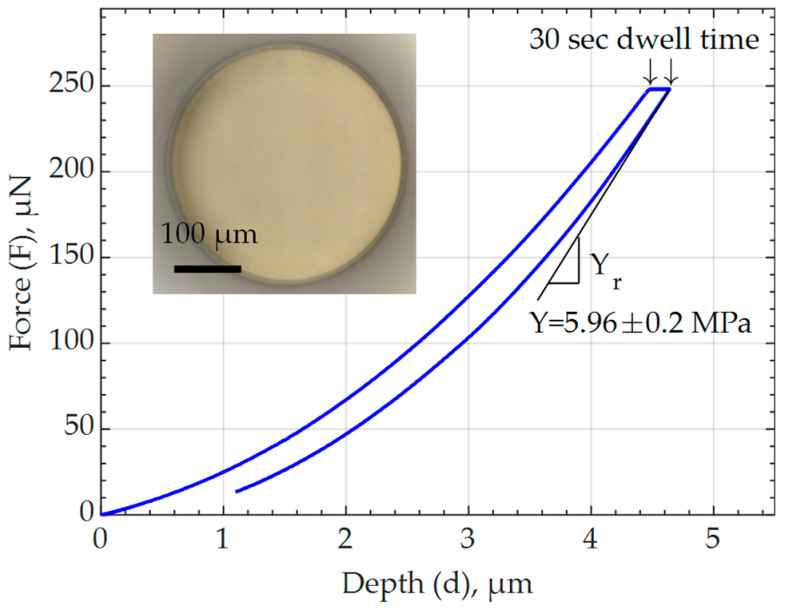
Nanoindentation results and tested 2PP-printed (cured) IP-PDMS cylindrical sample demonstrating the force–depth curve with the used dwell time and measured Young’s modulus.

**Figure 6 polymers-15-04377-f006:**
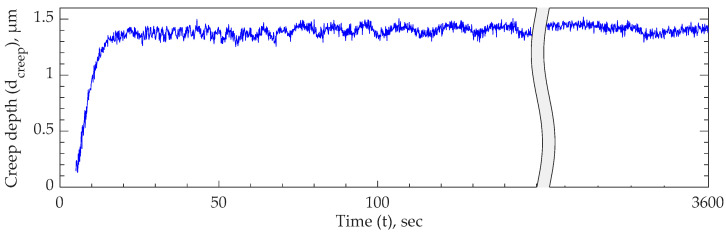
Creep test results demonstrating creep depth–time profile of 2PP-printed (cured) IP-PDMS tested through nanoindentation method.

**Figure 7 polymers-15-04377-f007:**
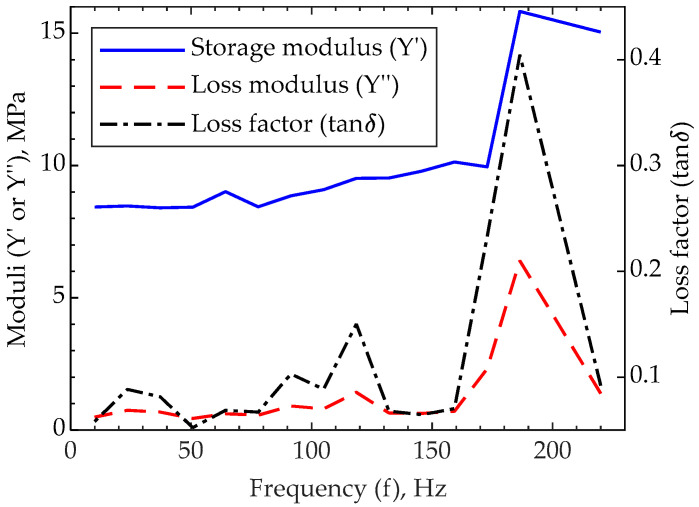
NanoDMA results demonstrating storage and loss moduli and loss factor of 2PP-printed (cured) IP-PDMS tested through nanoindentation method with 250 µN applied force.

**Figure 8 polymers-15-04377-f008:**
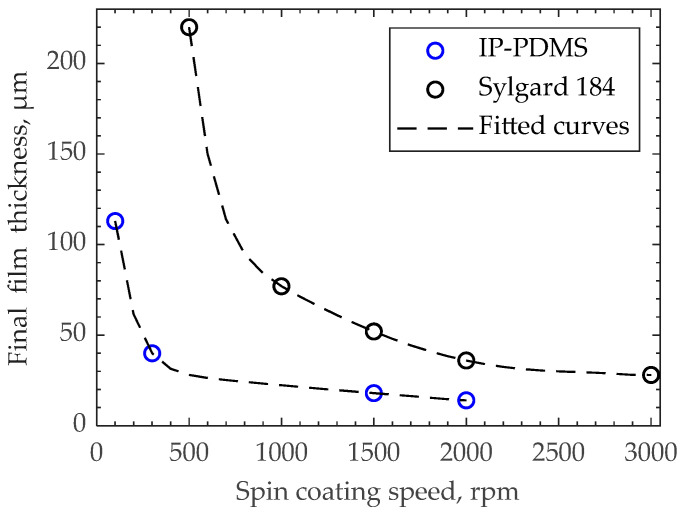
Final thickness of cured IP-PDMS films after spin coating at various speeds for 1 min on ITO-coated glass for dielectric and friction tests.

**Figure 9 polymers-15-04377-f009:**
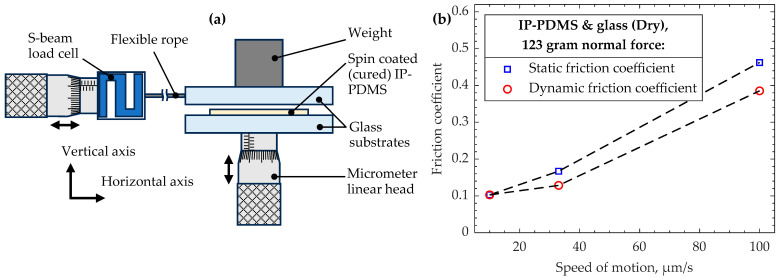
(**a**) Custom setup for measuring coefficients of friction between dry glass and spin-coated (cured) IP-PDMS and (**b**) measured results for various speeds of motion with 123 g of normal force.

**Figure 10 polymers-15-04377-f010:**
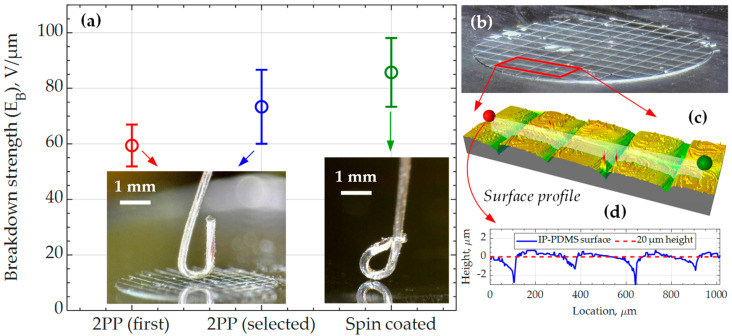
(**a**) Dielectric breakdown strength of 20 µm thick 2PP-printed (cured) and 18 µm thick spin-coated (cured) IP-PDMS samples at DC applied voltage. The measurements for 2PP samples are broken into two groups: the breakdowns that occurred first through the closest local defect like a bubble (first) and the following (selected) breakdowns that occurred through the visually integral part of the 2PP-printed sample. (**b**) Microscopy image of 3.35 mm diameter 2PP-printed and tested IP-PDMS cylinder with visible bubbles, (**c**) profilometry image of the portion of the cylinder with (**d**) evaluated surface profile.

**Table 1 polymers-15-04377-t001:** Printing parameters for IP-PDMS on ITO-coated glass substrate using 25x NA 0.8 lens.

Parameter	Value
Hatching Distance	0.3 µm
Slicing Distance	0.3 µm
Contour Distance	0.2 µm
Core Laser Power	80%/40 mW
Shell Laser Power	60%/30 mW
Core Scan Speed	100,000 µm/s
Shell Scan Speed	20,000 µm/s

**Table 2 polymers-15-04377-t002:** Summary of characterization performed in this paper on uncured IP-PDMS resin, 2PP-printed (cured), and spin-coated (cured) samples.

Property	Value				Figure
Transmittance (uncured IP-PDMS resin)	>90% for wavelengths above 440 nm <20% for wavelengths below 420 nm				Figure 3
Contact angle	11.5 ± 0.6° (uncured IP-PDMS resin on glass) 103.7 ± 0.5° (water on cured spin coated IP-PDMS)				Figure 4a Figure 4c
Surface tension (uncured IP-PDMS resin)	26.7 ± 4.2 mN/m				Figure 4b
Young’s modulus *	5.96 ± 0.2 MPa				Figure 5
Creep *	no secondary creep observed at room temperature				Figure 6
Storage modulus *	8.5–10 MPa @ 10–160 Hz		~15 MPa @ ≥170 Hz		Figure 7
Loss factor *	0.05–0.15 @ 10–160 Hz		0.4 @ 170 Hz		
Friction coefficient with glass (spin coated cured IP-PDMS)	@ 10 µm/s	@ 33 µm/s		@ 100 µm/s	Figure 9
Static	0.103	0.167		0.462	
Dynamic	0.103	0.129		0.385	
Relative dielectric permittivity (spin coated cured IP-PDMS)	2.63 ± 0.02 @ 10^1^–10^4^ Hz				-
Breakdown strength	first BD *†	selected BDs *†		spin-coated (cured) IP-PDMS	Figure 10
	59.4 ± 7.5 V/µm	73.3 ± 13.3 V/µm		85.7 ± 12.4 V/µm	

Plus/minus values represent standard deviation calculated utilizing Bessel’s correction. * 2PP-printed (cured) IP-PDMS samples were used. † ‘first BD’ and ‘selected BDs’ refer to the first breakdown on the film stimulated by a local defect and selected breakdown that occurred through the material, respectively.

## Data Availability

Data are contained within the article.

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
