# Peer review of "Characterization of Photocurable IP-PDMS for Soft Micro Systems Fabricated by Two-Photon Polymerization 3D Printing"

_polymers, 2023, doi:10.3390/polym15224377_

Round 1
Reviewer 1 Report
Comments and Suggestions for Authors
The manuscript by Govindarajan et al. provides characterization of the IP-PDMS material and values for some of its properties. While the approach and analysis are interesting to the readers of Polymers, the authors are asked to revise their manuscript and take into account the below list of comments and concerns.
In line 142, the authors mention “deionized water, acetone, and isopropanol” without mentioning the conductivity of water or the purity of the chemicals at the beginning of section 2. This information is required.
In line 143, the authors mention “blow dried with an air-ball”. What exactly do the authors mean by air-ball and how is the process accomplished?
In line 153, the authors mention “allowed to reach thermodynamic equilibrium”. What do the authors mean by this statement and how can this be repeatedly performed objectively and without operator bias?
The authors are also required to mention the volume of the sessile drop, what considerations they took into account when calculating this volume and how long did it take for the drop to reach “thermodynamic equilibrium”?
How different would the 3 graphs in Figure 2 be if the materials were uncured? Does the curing process have an effect on the absorbance levels of the different functional groups present? Also, does the curing process completely remove or add any specific functional group?
What is the color of uncured IP-PDMS resin used in this study and how does the transmittance of this material given in figure 3 compare to that of other types of commercially available PDMS (Sylgard 184 and 186)? The authors should include a picture of the uncured IP-PDMS used in this study and provide a comparison of the transmittance spectra of other PDMS in the text.
How would the graph in figure 3 look like as a function of curing time/percentage? Would the fully cured material allow the complete transmittance of this range of wavelengths? Will a partially cured material also have an almost 0% transmittance at 390 nm?
In section 3.3, the authors measured the contact angle and provide a value after the plus/minus sign. The authors have to mention how many samples were used in measuring the contact angle and what does the value after the plus/minus sign indicate. Is it the std. error and how was it calculated? This has to be included in the text.
The authors need to show “he” schematically in figure 4 (similar to figure 4a, the authors should draw a double headed arrow from a line at the free surface to another line at the bottom of the meniscus or clarify if they had measured it differently). How was “he” measured? Using ImageJ or using other techniques?
The authors provided the calculated surface tension and provide the “degree of error” after the plus/minus sign. How did the authors calculate that “degree of error” in surface tension? Did they perform a propagation of error analysis? This has to be included in the text. This also applies to the other properties that the authors calculate using equations such as Young’s modulus, permittivity, etc.
A picture of the contact angle measurement of water on IP PDMS needs to be included in the text. Additionally, the authors have to clarify the error calculation.
The authors dropped the Young’s modulus and Poisson’s ratio in equation 2 due to “the negligible compliance of the indenter tip compared to IP-PDMS”. How does the non-compliance characteristics of the indenter tip impact these values in equation 2? These are property values of the indenter tip that are irrespective of the application. Therefore, it’s recommended that the authors use these values and then compare the calculated Young’s modulus to that reported by the manufacturer.
The title of the y-axis of figure 8 should be corrected to just say “Final film thickness, um” by removing “IP-PDMS” since it also shows the final thickness of Sylgrad 184. The authors have to clarify what are the other test parameters in figure 8 such as amount of PDMS, temperature, curing time, etc.
A picture of the custom set up used to measure the friction coefficients should be included.
Finally, while the authors did a good job in characterizing IP-PDMS, I recommend that the authors improve the quality of the paper by adding more details and clarifications to provide a better understanding of the methods used in their investigations. This will make it easier on the reader to follow and would allow the independent replication of their results. With the above concerns, I can only recommend this manuscript for publication after a major revision.
Comments on the Quality of English LanguageThe authors are encouraged to proofread their manuscript. There are a couple of grammatical mistakes that can be avoided.
Reviewer 2 Report
Comments and Suggestions for Authors
In this study the authors report the characterization of relevant properties of a commercial two-photon curable polydimetylsiloxane (IP-PDMS) resin such as dielectric permittivity, young’s modulus, breakdown strength, creep, wettability, viscoelastic behavior. In light of such properties the material can be considered promising for applications in soft materials based technologies including energy storage, sensing actuation, etc. The study is interesting in the fields of soft materials and novela s it reports a full materials’ characterization for the first time. The study is presented clearly and the conlusions are supported by the results. In order to further improve the quality of the presentation I recommend to better specify when the tests are carried out on uncured IP-PDMS, cured IP-PDMS and printed vs spin coated form by renaming the different samples/materials with codes in graphs, captions and text, for example U-PDMS (uncured), C-PDMS (cured PDMS), P-PDMS (printed), SC-PDMS (spin coated). Moreover, in the abstract the authors should also specify what exact type of samples the properties thye list refer to.
Round 2
Reviewer 1 Report
Comments and Suggestions for Authors
The authors did a good job in improving their manuscript according to initial comments and recommendations provided earlier. However, the authors have to do the following corrections:
The value of the surface tension in the abstract has to be consistent with the calculated one provided in table 2. Authors are encouraged to check the other values to make sure they are consistent throughout the different sections of the manuscript.
The authors are encouraged to add their following reply statement “IP-PDMS does not have solvents in its composition and reasonably longer waiting time does not affect the measured contact angle, as we observed in the test.” In line 161 after “by waiting for one minute”.
The caption of Figure 9 needs to be fixed.
The authors are encouraged to check the formatting of the different items in the “Refences” section.
Comments on the Quality of English LanguageSome minor grammatical errors are present especially before the revised statements.
